# Relevance of Translation Initiation in Diffuse Glioma Biology and its Therapeutic Potential

**DOI:** 10.3390/cells8121542

**Published:** 2019-11-29

**Authors:** Digregorio Marina, Lombard Arnaud, Lumapat Paul Noel, Scholtes Felix, Rogister Bernard, Coppieters Natacha

**Affiliations:** 1Laboratory of Nervous System Disorders and Therapy, GIGA-Neurosciences Research Centre, University of Liège, 4000 Liège, Belgium; marina.digregorio@uliege.be (D.M.); alombard@chuliege.be (L.A.); pnlumapat@uliege.be (L.P.N.); felix.scholtes@uliege.be (S.F.); bernard.rogister@uliege.be (R.B.); 2Department of Neurosurgery, CHU of Liège, 4000 Liège, Belgium; 3Department of Neurology, CHU of Liège, 4000 Liège, Belgium

**Keywords:** gliomas, protein synthesis, translation, cap-dependent, IRES

## Abstract

Cancer cells are continually exposed to environmental stressors forcing them to adapt their protein production to survive. The translational machinery can be recruited by malignant cells to synthesize proteins required to promote their survival, even in times of high physiological and pathological stress. This phenomenon has been described in several cancers including in gliomas. Abnormal regulation of translation has encouraged the development of new therapeutics targeting the protein synthesis pathway. This approach could be meaningful for glioma given the fact that the median survival following diagnosis of the highest grade of glioma remains short despite current therapy. The identification of new targets for the development of novel therapeutics is therefore needed in order to improve this devastating overall survival rate. This review discusses current literature on translation in gliomas with a focus on the initiation step covering both the cap-dependent and cap-independent modes of initiation. The different translation initiation protagonists will be described in normal conditions and then in gliomas. In addition, their gene expression in gliomas will systematically be examined using two freely available datasets. Finally, we will discuss different pathways regulating translation initiation and current drugs targeting the translational machinery and their potential for the treatment of gliomas.

## 1. Introduction

Gliomas are the most frequent tumors of the central nervous system and include diffuse and circumscribed gliomas [1]. Diffuse gliomas are the most frequent cancer of the central nervous system in adults and will therefore be the focus of this review. In the latest classification for diffuse gliomas, the World Health Organization (WHO) has taken molecular characteristics into account as it becomes evident that histology on its own is not sufficient to characterize these brain tumors. For example, the presence of mutations in the isocitrate dehydrogenase 1 and 2 (*IDH1* and *IDH2*) and the association of complete 1p/19q co-deletion is now included in the classification of diffuse gliomas [1]. Grade II gliomas are low grade gliomas and include diffuse astrocytomas and oligodendrogliomas. Grade III gliomas include anaplastic astrocytomas and anaplastic oligodendrogliomas. Grade IV is the highest grade and includes glioblastomas also known as glioblastoma multiforme (GBM). GBM is the most aggressive glioma subtype and accounts for 60% of total gliomas [2,3,4]. Based on single-cell gene expression, GBM tumor cells can be further sorted into four states: oligodendrocyte-precursor-like, neural-progenitor-like, astrocyte-like, or mesenchymal-like [5]. Interestingly, these four states correspond to the four subtypes previously established by Verhaak et al. from The Cancer Genome Atlas Research (TCGA) whole-genome RNA sequencing study: Proneural, Neural, Classical, and Mesenchymal, respectively [6,7]. Unfortunately, so far, the Verhaak’s classification remains unhelpful for the choice of a specific treatment for a particular GBM subtype. Indeed, the median survival following GBM diagnosis remains short (ranging from 9 to 14 months) independently of the genetic subtype and despite standard therapy which combines surgical resection of the tumor followed by radiotherapy and chemotherapy using temozolomide (TMZ), the most commonly used chemotherapeutic agent [8]. The identification of new targets for the development of novel therapeutics are therefore needed in order to improve this devastating overall survival rate.

Cancer cells are highly proliferative, migratory and invasive. In addition, they are continually exposed to stressors such as hypoxia and nutrient deprivation, forcing them to constantly and rapidly adapt their protein production to survive. To do so, cancer cells can exploit their translational machinery to maintain cellular activity even in times of high physiological stress and can continue to synthesize proteins promoting their survival and necessary for their biology [9]. This phenomenon has been described in several cancers including human lung, breast, and prostate cancers and has been recently reviewed [10]. Thus, targeting the protein synthesis pathway has been postulated as an interesting new therapeutic avenue for several types of cancer [10,11]. However, the specific oncogenic role of translation in gliomas and in GBM, in particular, remains to be clarified. Translation is composed of four main phases, namely initiation, elongation, termination, and ribosome recycling. In this review, we will focus on translation initiation in gliomas, as this is the most critical and rate-limiting step of protein synthesis [12]. The different translation initiation protagonists will be described then their expression and roles in gliomas will be discussed. In addition, their gene expression in gliomas will systematically be examined using two freely available datasets: the REpository for Molecular BRAin Neoplasia DaTa (REMBRANDT; Affymetrix HG U133 v2.0 Plus; Control: *n* = 21, oligodendroglioma: *n* = 66, astrocytomas: *n* = 145, and GBM: *n* = 214) and data generated by The Cancer Genome Atlas Research Network (TCGA, https://www.cancer.gov/tcga, Affy Human Exon 1.0 ST; Control: *n* = 11, Classical: *n* = 54, Mesenchymal *n* = 58, Neural = 33, and Proneural *n* = 57 based on Verhaak’s classification) (Table 1 and Table 2) [6,13]. These two datasets were obtained through the independent Betastasis genomics analysis and visualization platform, and GraphPad Prism (version 5.03 for Windows, GraphPad Software, San Diego California USA, www.graphpad.com) was used for statistical analysis. Finally, we will discuss different pathways regulating translation initiation as well as current drugs targeting the translational machinery and their potential for the treatment of gliomas.

## 2. Overview on Translation Initiation

Two distinct initiation processes exist in eukaryotic cells: cap-dependent and cap-independent initiation [14]. Cap-dependent translation can be divided into two major steps (Figure 1). During the first step, the eukaryotic translation initiation factor (eIF) 4F (eIF4F) complex comprised of eIF4E, eIF4G, and eIF4A, assembles on the 7-methylguanosine 5′-triphosphate (m7GTP) cap structure present on the 5’ end of the mRNA (Figure 1a). During the second step of cap-dependent translation, the small ribosomal subunit (40S) associated with eIF1, eIF1A, eIF3, and eIF5 interacts with the ternary complex constituted of eIF2, a switch-protein bound to GTP and the initiator methionyl-transfer RNA (Met-tRNAi). Together they form the 43S preinitiation complex (43S PIC) (Figure 1b). The 43S PIC then recognizes and binds the eIF4F complex near the cap to create the 48S PIC (Figure 1c). The 40S ribosomal subunit associated with eIFs uses ATP hydrolysis as energy supply to scan mRNA from the 5’ untranslated region (UTR) until it reaches the initiation codon (AUG), allowing the localization of the Met-tRNAi in the future peptidyl site, or P site, of the ribosome. The 60S ribosomal subunit is then recruited and joins the 40S ribosomal subunit which triggers eIF2-GTP hydrolysis, leading to eIFs release and the formation of the 80S initiation complex (Figure 1d) [15]. At this stage, the initiation complex is fastened on the mRNA and the elongation begins by accepting a second tRNA with its amino acid in the aminoacyl site (A site) of the ribosome [16].

Whereas translation of most eukaryotic mRNAs depends on the presence of a m7GTP cap on the 5’ UTR end, there is now evidence that it is not the only mode of initiation. Indeed, in the absence of cap-binding machinery or when cap-dependent initiation has been inhibited following cellular stress, alternative mechanisms can be initiated to ensure that proteins required for cell survival remain expressed. Several modes of translation have been proposed to explain cap-independent translation of mammalian mRNA with internal ribosomal entry site (IRES) being the most studied so far [17,18]. IRES-dependent initiation and current research in gliomas will be discussed hereafter.

## 3. Cap-Dependent Initiation

Translation initiation includes several steps and involves multiple players. We will attempt to facilitate the comprehension of this rather complex biological process by describing and discussing intermediary steps as depicted in Figure 1. We therefore encourage readers to use Figure 1 as a visual support.

### 3.1. The eIF4F Initiation Complex

The eIF4F complex includes three subunits, each with their specific role in the initiation process: (i) eIF4E binds m7GTP at the 5′ UTRs of mRNAs and is the less abundant eIF making it a limiting factor of translation, (ii) eIF4G is a scaffold protein whose role is to coordinate binding of ribosomal subunits with other eIFs and (iii) eIF4A is a RNA helicase ATP-dependent protein which unwinds mRNA during translation [10].

#### 3.1.1. eIF4E

eIF4E is a limiting factor and thus a key element during protein synthesis due to its low abundance and its regulation through phosphorylation and its interaction with other proteins [19,20,21]. The eIF4E family comprises three members; eIF4E1, eIF4E2 also called 4EHP, and eIF4E3, all of which have the capacity to bind mRNA caps but with different affinity [22]. eIF4E1 is considered as the canonical eIF4E and binds m7GTP with the highest affinity [22]. In the literature, eIF4E1 is the most studied. However, the distinction between these three members is often unclear. Hereby, we will clarify which member has been studied, when possible, or use eIF4E for investigations which did not specify which isoform was considered.

In 2005, Gu et al. reported that eIF4E protein was overexpressed in astrocytes, proliferative endothelial cells, and in vascular endothelial cells present in human GBM tissue samples [23]. This conclusion was based on immunohistochemistry performed on paraffin embedded tissue from five human GBM cases. Their protocol included a primary antibody incubation step of 25 minutes which is rather short to allow sufficient time for a typical antibody-antigen interaction. Furthermore, no co-labelling was performed and haematoxylin-eosin counterstaining alone does not permit clear conclusion on the cell types expressing eIF4E [23]. Nevertheless, higher expression of EIF4E in GBM was later confirmed by another team. They used a tissue microarray including cores of GBM (*n* = 25) and control brains (*n* = 50) combined with assisted quantitative scoring of the immunostaining [20]. Levels of eIF4E and its phosphorylated form (p-eIF4E) increase with glioma tumor grade and are predictive of poor survival [24]. Phosphorylation of eIF4E on Ser209 is regulated by mitogen-activated protein kinase (MAPK) interacting protein kinases (MNKs) and occurs once eIF4E is bound to the m7GTP cap structure and has been stabilized by eIF4G (Figure 2). Phosphorylation of eIF4E modifies the repertoire of translated mRNA with increased translation of mRNAs encoding pro-tumorigenic factors such as matrix metalloproteinases or vascular endothelial growth factor C [21]. Interestingly, expression of p-eIF4E is highly sensitive for the diagnosis of GBM (94.3%) and to a lesser extent for the differentiation between high-grade and low-grade astrocytomas (81.4%). Furthermore, the combination of elevated levels of p-eIF4E and eIF4E-binding protein 1 (4E-BP1, an inhibitor of eIF4E discussed below) has a positive predictive value of 100% for GBM diagnosis and was therefore suggested as a diagnostic tool for GBM in small biopsy materials [24]. In line with this finding, it was shown that lower levels of eIF4E phosphorylation, together with alteration of other proteins including p-Akt, could act as anti-angiogenic factors in glioma cells [25].

Analysis of data available on REMBRANDT and TCGA databases revealed that *EIF4E1* and *EIF4E3* mRNA levels are downregulated whereas *EIF4E2* is overexpressed in gliomas (Table 1). Interestingly, contrary to eIF4E1, eIF4E2 has lower affinity for the m7GTP cap and more importantly, might be a repressor of translation [22]. Thus, decreased expression of eIF4E1 and eIF4E3 together with increased eIF4E2 expression suggests a global repression of cap-dependent translation in gliomas. Although data obtained from REMBRANDT and TCGA databases need to be validated at protein levels, this observation illustrates the importance of distinguishing members of the eIF4E protein family when studying them in gliomas.

The importance of making the distinction between eIF4E family members is also illustrated by a potential role for eIF4E2 in GBM. Indeed, eIF4E2 is required for tumor progression and controls 3’ UTR hypoxia response element-containing mRNA translation. Inhibition of eIF4E2 under hypoxic condition decreases wound healing, cell migration, and sphere formation in the U87MG human glioma cell line [26]. These interesting findings should be confirmed in primary GBM cells. Additional in vitro and in vivo studies support a role for eIF4E in gliomas. For example, eIF4E is involved in the regulation of genes coding for proto-oncogene proteins like c-myc and cyclin D1 which are involved in cell proliferation and promote tumor growth [27,28,29,30,31]. This potential role of eIF4E proteins in tumor biology is also supported by studies confirming that eIF4E is involved in cell proliferation and tumor growth [32,33]. Ribavirin, an anti-viral drug that disrupts the interaction between eIF4E and the 5’ UTR end of mRNA (Figure 2), inhibits GBM cell growth and migration, and increases radio-/chemo-therapy efficacy in vivo [33,34]. In agreement with this observation, the overexpression of eIF4E in U373 cells, a human GBM astrocytoma cell line, increases cell proliferation under normoxic conditions and overall tumor size in a xenograft model. Contrary to the study showing that inhibition of eIF4E increases radiosensitivity, Rouschop et al. showed that eIF4E overexpression induces a decrease in the number of hypoxic cells and increases radiosensitivity [32]. Furthermore, a research group investigated the effect of hypoxia and serum deprivation, two conditions often present in the tumor microenvironment [9], on the proteome of U87MG cells and found that eIF4E was overexpressed under these conditions [35]. Finally, eIF4E was suggested to play a role in the maintenance and self-renewal capacity of GBM initiating cells (GICs) through Sox2 regulation [36]. GICs infiltrate the parenchyma surrounding the tumor mass and consequently escape surgical resection. In addition, GICs are resistant to radio- and chemo-therapy, making them strong suspect in GBM recurrence and therefore, an ideal target to prevent those relapses [37,38].

As eIF4E expression is low under basal conditions and is a limiting factor of cap-dependent translation initiation, it is important to consider factors regulating its availability. One of the two known mechanisms involved in the regulation of eIF4E availability is the eukaryotic translation initiation factor 4E nuclear import factor 1 (*EIF4ENIF1* also known as 4E-T), which regulates eIF4E trafficking to the nucleus [39]. Depletion of eIF4E from the cytoplasm would ultimately result in the suppression of cap-dependent translation. The presence of eIF4E in the nucleus suggests additional functions beside its role in translation. One potential role for eIF4E in the nucleus is the regulation of mRNA trafficking outside the nucleus [39]. Interestingly, whereas eIF4E is present in the cytoplasm and in the nucleus in normal cells, the majority of eIF4E is nuclear in cancer cells [39]. In addition to its role in the processes responsible for the subcellular localization of eIF4E, 4E-T also participates in mRNA turnover through its direct interaction with eIF4E and components of the mRNA decay machinery [40]. According to our analysis shown in Table 1, the expression of *EIF4ENIF1* is significantly decreased in astrocytomas and all GBM subtypes suggesting a potential decrease in mRNA recycling in gliomas and/or an alteration in eIF4E subcellular localization. This, of course, requires further investigation and validation. Other major inhibitors of eIF4E include 4E-BPs which are small proteins competing for eIF4G, the direct partner of eIF4E in the eIF4F initiation complex. Therefore, 4E-BPs will be discussed in the next section.

#### 3.1.2. eIF4G

Among the eIF4G family members, only eIF4G1 and eIF4G3 (also known as eIF4GI and eIF4GII, respectively) are directly involved in the cap-dependent translation, while eIF4G2 (also known as Dap5) has been found to play a role in cap-independent translation and will be discussed in the corresponding section [41,42]. eIF4Gs are scaffold proteins which means that they interact with RNA as well as many other proteins including eIF4E, poly(A) binding proteins (PABPs), eIF3, eIF4A, and MNK1 (the kinase able to phosphorylate eIF4E, see above) [41,43]. Thus, they participate in the eIF4F complex formation during cap-dependent translation (Figure 1). PABP is an RNA-binding protein which links eIF4G with the 3′ poly(A) tail of eukaryotic mRNAs, thus circularizing mRNAs. PABP contributes to translation initiation potentially by several mechanisms including enhancing eIF4F affinity for the cap structure, increasing eIF4F activity or promoting ribosome recycling [44]. The expression of eIF4G proteins is altered in different types of human cancer including in gliomas as shown in our REMBRANDT and TCGA data analysis (Table 1) [45]. These analyses reveal indeed that the expression of *EIF4G1* and *EIF4G2* is significantly increased in gliomas whereas *EIF4G3* is decreased in gliomas and all GBM subtypes.

Direct interaction between eIF4G and eIF4E seems to be a critical step in the initiation of protein synthesis. Members of the 4E-BP family, namely 4E-BP1, 4E-BP2, and 4E-BP3, bind and thus sequester eIF4E which prevents its interaction with eIF4G and in turn blocks translation (Figure 1a). Phosphorylation of 4E-BPs triggers eIF4E release and consequently permits eIF4E-eIF4G interaction and translation initiation [46]. Pathways involved in the regulation of 4E-BP phosphorylation will be discussed below in the corresponding section. Whereas *EIF4EBP1* and *EIF4EBP3* seem to be overexpressed in gliomas, this is less clear for *EIF4EBP2* as data from REMBRANDT show an increase in mRNA levels in oligodendrogliomas and astrocytomas, but not in GBM, whereas, according to the TCGA analysis, there is a decrease in *EIF4EBP2* in most GBM subtypes (Table 1). In agreement, Martínez-Sáez et al. used immunohistochemistry to investigate 4E-BP1 in 104 diffuse infiltrating astrocytomas including diffuse astrocytomas (grade II, *n* = 19), anaplastic astrocytomas (grade III, *n* = 25), and GBM (grade IV, *n* = 60). They found that levels of 4E-BP1 and p-4E-BP1 correlate with tumor grades, with the highest level detected in GBM. However, no significant difference in survival was observed between the different groups [24]. Disruption of eIF4F complex using a 4E-BP mimetic molecule (4EG-1) triggers apoptosis in GBM cells and reduces tumor growth in a xenograft model (Figure 2) [47,48]. In line with this, Dubois et al. demonstrated that 4E-BP1 silencing accelerates the growth of xenografted U87MG cells in mice [49]. Results obtained from these studies are surprising since 4E-BP1 is increased in gliomas, which suggests that higher expression of 4E-BP1 facilitates the development of the disease [24].

#### 3.1.3. eIF4A

The last subunit belonging to the eIF4F initiation complex is the RNA helicase eIF4A, which unwinds mRNA secondary structures and permits the recruitment of the 43S PIC (Figure 1). In addition, eIF4A is also required to untwist mRNA during scanning, leading to recognition of the initiation codon then assembly of the 60S subunit and the beginning of the elongation step. Three eIF4A isoforms, eIF4A1-A3, have been identified in mammals and their structures and functions have been reviewed [50,51]. eIF4A1 and eIF4A2 take part in the eIF4F complex while eIF4A3 is involved in the nonsense-mediated decay machinery, a mechanism used by eukaryotes to eliminate mRNA transcripts which contain premature stop codons or nonsense mutations. eIF4A1 is highly expressed in actively dividing cells and is upregulated in various cancers. Conversely, eIF4A2 is present at high levels in resting cells and low levels of eIF4A2 has been associated with poor outcome for patients with breast and non-small-cell lung cancers [52,53]. In agreement with this finding, our analysis shows that *EIF4A1* and *EIF4A3* are significantly increased in gliomas including GBM for the four subtypes compared to non-cancerous brain tissue whereas *EIF4A2* expression is significantly downregulated (Table 1). To the best of our knowledge, eIF4A proteins have not yet been investigated in human GBM samples by other teams and further studies should thus aim at confirming these data.

The RNA helicase activity of eIF4A1 and eIF4A2 as well as their interaction with eIF4G can be inhibited by programmed cell death 4 (PDCD4). PDCD4 prevents eIF4F complex assembly and consequently inhibits cap-dependent translation by binding eIF4A (Figure 1a). PDCD4 phosphorylation triggers its degradation which frees up eIF4A thus allowing eIF4F complex formation [54]. PDCD4 acts as tumor suppressor and is often decreased in cancer [55,56]. In 2007, Gao et al. studied PDCD4 at mRNA and protein levels in 30 glioma samples from grade I-II to grade IV. Using RT-PCR, western blot and immunocytochemistry, they found that PDCD4 was decreased in gliomas [57]. These findings were confirmed by our analysis of TCGA dataset but only for the classical GBM subtype. PDCD4 expression can be regulated at various levels including by miRNAs [58], some of which have been reported to be upregulated in GBM [59,60]. Inhibition of PDCD4, and consequently release of eIF4A and activation of translation, decreases apoptosis, promotes cell cycle arrest at G0/G1, and stimulates glioma stem cell proliferation as well as GBM cell invasiveness. Furthermore, enhanced expression of PDCD4 decreases tumorigenic capacity of glioma stem cells in vivo as demonstrated in a GBM xenograft mouse model [60].

Natural compounds including pateamine have also been shown to interfere with eIF4G and eIF4A interaction leading to a lower number of mRNA-associated ribosomes and subsequently to inhibition of translation initiation. However, pateamine has not yet been investigated in glioma cells and subsequent biological impacts on gliomas as well as other cancer types remain to be studied [61,62].

### 3.2. The 43S Preinitiation Complex (PIC): eIF1, eIF1A, eIF3, and eIF5 

Following eIF4F complex formation on the m7GTP cap structure at the 5′ UTR end of an mRNA, the 43S PIC is put into place. During this second step of translation initiation, eIF3 together with eIF1, a key factor in the initiation codon recognition [63], eIF1A and eIF5 associate with the 40S ribosomal subunit and the ternary complex to form the 43S PIC (Figure 1b) [64,65,66,67,68].

#### 3.2.1. eIF3

eIF3 is composed of 13 subunits in human and among these, five of them have been shown to be overexpressed in different cancer types, namely eIF3A, B, C, D, and E [69,70,71,72,73]. According to our REMBRANDT data analysis, all EIF3 subunits, except subunit K, are upregulated in gliomas (however not always in all glioma subtypes) relative to non-tumoral brain tissue. When considering the different GBM subtypes separately using the TCGA dataset, it appears that specific eIF3 subunits are preferentially overexpressed in different genetic subtypes. For example, *EIF3B, D, G,* and *I* subunits are significantly overexpressed in classical and mesenchymal subtypes, the later having the worst prognosis. Unfortunately, REMBRANDT and TCGA databases on Betastasis do not encompass the expression of *EIF3C* (Table 1). Nevertheless, eIF3C together with other eIF3 subunits have been studied in glioma tissues, in vitro and in vivo by other research teams [74,75,76,77,78].

In 2012, Liang et al. were the first to find evidence that eIF3 subunits could play a role in glioma. They investigated *EIF3B* expression by RT-PCR in gliomas (*n* = 10) ranging from grade I to grade IV and found no significant difference between tumor grades. However, they found that reduction of eIF3B expression decreases cell proliferation, induces cell cycle arrest and triggers apoptosis in U87MG cells [76]. In a later study, the same group found that eIF3D is increased in gliomas (*n* = 35) and that its inhibition in U87MG cells induces similar effects as what they obtained for eIF3B [77]. Sesen et al. (2014) studied eIF3E in human GBM cell lines and demonstrated that blocking its expression decreases GBM cell proliferation, blocks cell cycle and increases apoptosis [78].

In addition, eIF3C (which expression data are lacking in REMBRANDT and TCGA as evoked above) was investigated by immunohistochemistry in glioma samples (*n* = 83). This study showed a significant increase in eIF3C staining in glioma samples compared to staining detected in brain tissues from traumatic brain injury patients used as control (*n* = 25). Increased levels of eIF3C positively correlated with tumor grades. Furthermore, it was demonstrated that inhibition of eIF3C in vitro using siRNA reduces cell proliferation, decreases colony formation, increases apoptosis and promotes cell cycle arrest in U87MG cells. Finally, these authors showed that lower eIF3C expression reduces tumor growth in vivo using a glioma xenograft mouse model [75].

Altogether, these studies corroborate our REMBRANDT and TCGA analyses and all support a role for eIF3 subunits in major biological processes and their potential involvement in gliomas. A study published during the preparation of this manuscript investigated all *EIF3* subunits expression in glioma using the Chinese Glioma Genome Atlas (CGGA, *n* = 272) together with data from TCGA (*n* = 595). Contrary to our findings showing that only specific *EIF3* subunits were overexpressed in all glioma subtypes, Chai et al. reported a significant increase for all subunits in glioma. The reason for this discrepancy is unclear but might be explained by the type of analysis used or the higher number of cases they analyzed [74]. Nevertheless, it can be concluded that eIF3 subunits are altered in gliomas and are involved in GBM cell proliferation. This conclusion is not surprising since eIF3 directly binds mRNA coding for proteins involved in the regulation of cancer cell growth including cell cycle arrest and apoptosis [79].

#### 3.2.2. eIF1, eIF1A, eIF5, and eIF5B

eIF1 and eIF1A are two small peptides which act synergistically to induce and maintain a conformational change in the 40S ribosomal subunit and promote an open conformation permitting efficient mRNA scanning. Once the 43S PIC scanning complex recognizes the start codon and the Met-tRNAi anticodon associates with it, eIF1 is released and eIF2-GTP is converted to eIF2-GDP with the help of eIF5, the eIF2 GTPase activating protein (Figure 1). Ejection of eIF1 from 40S induces a switch from an open to closed conformation thus probably ending the scanning step [80]. Finally, eIF5B together with eIF1A triggers the dissociation of most remaining eIFs and facilitates the assembly of the 60S ribosomal subunit to form the 80S complex [80,81].

The expression of eIF1 and eIF1A have not yet been studied in gliomas. Here, we found no change in mRNA levels between control brain tissue and any of the three gliomas subtypes (Table 1). However, as shown in the analysis of the REMBRANDT dataset, eIF1A domain-containing protein (*EIF1AD*) was significantly overexpressed in oligodendrogliomas and GBM. eIF1AD is an analogue of eIF1A. The role of eIF1AD is still unclear but its analogy with eIF1A suggests its involvement in ribosome biogenesis or protein synthesis [81]. The fact that *eIF1AD* but not eIF1 or eIF1A, is abnormally expressed in gliomas should encourage studies aiming at elucidating its function, in particular in gliomas.

Finally, we have analyzed *EIF5* and *EIF5B* expression in gliomas and found that *EIF5* is downregulated in astrocytomas and in GBM, only for the mesenchymal and proneural subtypes (Table 1). In addition to its role in the cap-dependent translation, eIF5B can act as ITAFs and is thus involved in the IRES-dependent initiation. So far, the involvement of eIF5B in gliomas has only been studied in the context of IRES-dependent translation and will therefore be discussed in the cap-independent section [82].

### 3.3. Additional eIFs: eIF4B, eIF4H, and eIF6

In addition to eIFs taking part in the eIF4F and 43S PIC complexes, crucial adjunct eIFs including eIF4B, eIF4H, and eIF6 also coordinate translation initiation.

#### 3.3.1. eIF4B/eIF4H

eIF4B and its paralogue eIF4H are chaperone proteins able to bind mRNA and eIF4A to stabilize unwound mRNA and stimulate eIF4A RNase and helicase activities [83,84]. Current literature on the role of eIF4B and eIF4H suggests a role in cell survival, cell proliferation, resistance to chemotherapy, and cell migration [85,86,87]. The expression and the role of eIF4B or eIF4H in gliomas have not yet been studied in details; however, the REMBRANDT data analysis shows a significant increase in *EIF4B* expression in oligodendrogliomas whereas no alteration was observed for neither of the two paralogues in GBM and astrocytomas compared to non-tumoral tissues. Though when GBM subtypes are considered separately using the TCGA database, *EIF4H* is downregulated in GBM but only in the proneural subtype (Table 1). As eIF4B activity is regulated at least partially by phosphorylation including on Ser422 by the S6-kinase 1 (S6K1) (Figure 2) [87,88], phosphorylation state of eIF4B and eIF4H should also be considered when studying gliomas.

#### 3.3.2. eIF6

Thanks to its double localization in the nucleus and in the cytoplasm, eIF6 takes part in ribosome biogenesis and maturation, respectively [89]. eIF6, also known as p27(BBP) or β4 integrin interactor, binds the 60S ribosomal subunit in the nucleus and, once in the cytoplasm, dissociates from 60S which leads to formation of the 80S complex and subsequent protein synthesis (Figure 1c,d) [90]. Its implication in human cancers has recently been reviewed by Zhu W. and colleagues [89]; however, its function and expression in gliomas remain unknown, with a single study published in 2014 by Saito K. et al. which demonstrated an indirect role for eIF6 in ribosomal biogenesis using U87MG cells. In fact, impaired expression of elongation factor Tu-GTP binding domain containing protein 1 (*EFTUD1*) involved in ribosome biogenesis, delocalizes eIF6 from the nucleus to the cytoplasm in U87MG cells. *EFTUD1* being overexpressed in gliomas might alter eIF6 subcellular localization and consequently prevent eIF6 from performing its functions [91]. Supporting a role for eIF6 in gliomas, we found that *EIF6* is overexpressed in gliomas including in GBM for all subtypes except the proneural (Table 1).

### 3.4. Ternary Complex (eIF2-GTP and the Initiator Methionyl-Transfer RNA)

eIF2 is a switch protein that associates with GTP and Met-RNAi to form the Ternary complex. This complex then binds the 40S ribosomal subunit together with several eIFs to give rise to the 43S PIC which will first associate with the eIF4F complex then scan the 5′ end of mRNA until it reaches the initiation codon (Figure 1b,c). Upon association of the 40S with the 60S ribosomal subunit, eIF2 latent GTPase enzymatic activity is released and eIF2-bound GTP is converted to GDP. This is the signal for eIFs to dissociate from the complex, ending the translation initiation step, and for protein synthesis to start with the elongation phase (Figure 1d) [15]. Once in its eIF2-GDP form, eIF2 is bound to eIF5 and is no longer active. eIF2B is then required to induce the dissociation of eIF2-GDP from eIF5 and for the catalytic exchange of GDP to GTP (Figure 1d) [92,93]. eIF2B comprises five subunits annotated α–ε and coded by *EIF2B1-5* genes respectively. The α, β, and δ subunits recognize phosphorylated eIF2α and modulate eIF2B activity by regulating the catalytic subunits γ and ε [94,95,96]. The catalytic subunit ε is overexpressed in different human cancers and is involved in tumorigenicity [97]. The non-catalytic subunits have been less studied in cancers and none of the eIF2B subunits were investigated in gliomas. In our analysis of gene expression, all *EIF2B* subunits except *EIF2B3* (encoding the γ subunit) are overexpressed in gliomas compared to non-tumoral brain tissue. On the contrary, the *EIF2B3* subunit is down-regulated in gliomas including in GBM, but only in the classical subtype (Table 1).

eIF2 is a heterotrimeric structure composed of three subunits (α, β, and γ) coded by three distinct genes *EIF2S1–S3*, respectively [98]. *EIF2S1* is not significantly differentially expressed in gliomas compared to non-tumoral brain tissues nor in GBM when the different subtypes are considered individually. *EIF2S2* and *EIF2S3*, on the other hand, are both upregulated in gliomas with the γ isoform being the only subunit to be overexpressed in the mesenchymal GBM subtype compared to non-tumoral tissue (Table 1).

Phosphorylation of the eIF2α subunit plays a major role in the regulation of mRNA translation as it blocks the catalytic exchange of GDP to GTP catalyzed by eIF2B and, as a result, triggers the arrest of translation initiation [99]. It is therefore crucial to also consider eIF2α phosphorylation status. Accordingly, some of the pathways involved in the regulation of eIF2α phosphorylation and known to be disturbed in gliomas will be discussed below.

Four kinases are able to phosphorylate eIF2α: HRI, PKR, PERK, and GCN2 encoded by *EIF2AK1*, *EIF2AK2*, *EIF2AK3,* and *EIF2AK4*, respectively. During cellular stress, these kinases phosphorylate eIF2α on Ser51 to decrease global mRNA translation and allow cell adaptation to environmental conditions. These kinases are not activated by the same stress. HRI is activated by heme deficiency, PKR by viral infection, PERK by endoplasmic reticulum (ER) stress and GCN2 by decreased essential amino acids [100]. According to the REMBRANDT database, *EIF2AK1* is significantly overexpressed in gliomas compared to non-tumoral brain tissues (Table 1) which is in agreement with a TCGA analysis from Haapa-Pananen S. et al. in 2013. However, the latter considered all GBM subtypes together and performed a student *t*-test. In our TCGA analysis, the four GBM subtypes were considered separately and analyzed by a one-way analysis of variance (ANOVA) test which revealed no significant change in *EIF2AK1* expression. Nevertheless, blocking *HRI* expression using miRNA decreased proliferation and increased apoptosis in GBM cells [101]. PKR, the second eIF2α kinase, is generally activated by dsRNA from virus infecting cells. Once PKR is activated, it inhibits cell growth and protein synthesis to allow cell adaptation to environmental conditions [100]. This activation could therefore be used to inhibit protein synthesis and subsequently cell growth in glioma cells. Shir A. et al. have indeed demonstrated that, when complementary RNA specific to RNA from GBM cells is injected in these cells, dsRNA can be reconstituted to activate PKR. This activation induces GBM cell apoptosis in vitro and in vivo [102,103]. As activated PKR inhibits cell growth, it is surprising to note that *EIF2AK2* expression is significantly higher in gliomas compared to non-tumoral brain tissues (Table 1). However, as PKR activation is necessary for eIF2α phosphorylation, it is possible that assessing the expression of its gene *EIF2AK2* is not sufficient to assess its function in gliomas. *EIF2AK3*, the gene coding for PERK, is overexpressed in GBM compared to non-tumoral brain tissue but not in the other two gliomas subtypes. Also, surprisingly, when GBM subtypes are considered separately using the TCGA dataset, this significance is lost (Table 1). The implication of PERK in glioma cell biology will be described in the ER stress section. Finally, decrease in levels of essential amino acids activates GCN2, leading to eIF2α phosphorylation and activation of activating transcription factor 4 (ATF4). ATF4 is a transcription factor able to modulate amino acid response elements (AAREs)-containing genes [104]. In LN229 GBM cells, tryptophan (Trp) depletion activates the GCN2-p-eIF2α-ATF4 pathway but does not decrease protein synthesis. This pathway activation leads to tryptophanyl-tRNA synthetase overexpression. This last enzyme increases Trp incorporation in cancer cell proteins despite a lack of this amino acid in the environment. GCN2-p-eIF2α-ATF4 pathway could therefore maintain protein synthesis despite a decrease in essential amino acids in the environment [105]. Interestingly, we found that *EIF2AK4* (coding for GCN2) is increased only in GBM compared to non-tumoral brain tissues (Table 1). As for the other three kinases involved in eIF2α phosphorylation, our analysis of the TCGA database shows that *EIF2AK4* is not differentially expressed in GBM when the four subtypes are considered separately (Table 1).

The ER stress response is a good illustration of the role of eIF2α phosphorylation in the regulation of protein synthesis. Environmental stressors are prompt to trigger ER stress during which unfolded or misfolded proteins accumulate in the ER lumen thereby triggering an adaptive response called the “unfolded protein response” (UPR) [9,106]. During the UPR, cells under stress stop protein synthesis, attempt to refold proteins, and eventually trigger apoptosis if ER stress is too intense or persistent. Cancer cells, however, manage to survive in challenging environmental conditions particularly thanks to the UPR pathway in which eIF2α is implicated. The role of the UPR in cancer and in GBM has recently been reviewed [9,106]. We encourage readers to refer to Figure 1 published in the review from Obacz et al. (2017) to obtain a good illustration of the players involved in the UPR pathway [9]. During the UPR, the glucose regulated protein 78KDa (GRP78 or BIP) expressed in the ER lumen, binds to misfolded or unfolded proteins and dissociates from transmembrane ER stress sensors such as activating transcriptional factor 6 (ATF6), inositol-requiring enzyme 1 (IRE1) and protein kinase-like ER kinase (PERK). Only the role of PERK will be described in this review as it is the only UPR actor controlling mRNA translation. After GRP78 dissociation from PERK and its activation upon phosphorylation, eIF2α gets phosphorylated on Ser51 leading to repression of protein synthesis [107]. PERK also controls translation of the transcription factor ATF4 in order to modulate the expression of foldase/chaperon and autophagy genes but also cytidine-cytidineadenosine-adenosine-thymidine (CCAAT)/enhancer binding homologous protein (CHOP) expression to control cell apoptosis [9,108].

In glioma cells, ER stress triggered by different drug treatments increases p-eIF2α in conjunction with elevated expression of other ER stress transducers (i.e. p-ERK, ATF6, p-IRE-1, GRP78, CHOP, XBP-1) enhancing glioma cell death [109,110,111]. Interestingly, ER stress can also induce glioma cell autophagy through eIF2α and decrease tumor cell survival [112,113,114,115]. However, inhibition of autophagy with various drug treatments decreases p-4E-BP1, p-70S6K1, and the ribosomal S6 protein and increases expression of CHOP and p-eIF2α levels leading to glioma cell death. These last observations therefore suggest that autophagy could have a cytoprotective or cytotoxic effect in glioma cells through various ER stress transducers [112,116,117].

As already and briefly mentioned, another challenge in the therapeutic strategy for GBM is constitutively variable resistance to TMZ, which can develop after or during treatment. DNA alkylation damage induced by TMZ can be repaired by the DNA repair enzyme O6-methylguanine methyltransferase (MGMT) which can be overexpressed in GBM and cause TMZ resistance. MGMT inhibitors have therefore been studied in order to overcome TMZ chemoresistance. Bortezomib, a Food and Drug Administration (FDA), and European Medicines Agency (EMA) approved proteasome inhibitor used for the treatment of multiple myeloma, stabilizes poly-ubiquitinylated proteins, and triggers cell cycle arrest and cell death [118]. Interestingly, Bortezomib decreases NFκB and MGMT mRNA amounts, but can also induce eIF2α phosphorylation to decrease protein synthesis of MGMT and therefore acts as translational repressor of MGMT. These two effects of Bortezomib lead to inhibition of DNA repair in T98G GBM cells and is therefore expected to sensitize cells to TMZ [119]. However, through eIF2α phosphorylation, Bortezomib also induces stress granules formation and increases GBM cell resistance to death signals [120]. Despite promising effects in cultured GBM cells, Bortezomib did not show sufficient efficacy in GBM patients during phase II clinical trials [121]. This was further supported by others using in vitro experiments showing that Bortezomib also increases Akt and 4E-BP1 phosphorylation in GBM cell lines leading to cell division [122]. TMZ also acts as an ER stress inducer leading to dissociation of PERK from prolyl 4-hydroxylase beta polypeptide (P4HB), a chaperone protein implicated in ER stress. PERK thereby phosphorylates eIF2α and triggers GBM cell apoptosis due to protein synthesis arrest. One explanation for GBM cell resistance to TMZ is P4HB overexpression found in recurrent and TMZ-resistant GBM tumors. P4HB suppression with siRNA or bacitracin therefore bypasses TMZ resistance and sensitizes GBM cells to TMZ-induced ER stress [123]. Finally, TMZ resistance has also been linked with cap-independent translation and this will be discussed in the appropriate section below [28,124,125].

#### 3.4.1. PI3K/Akt/mTOR, MAPK/MNK, and AMPK Regulating Pathways

A number of studies have demonstrated that key pathways which regulate major metabolic functions, such as PI3K/Akt/mTOR, MAPK/MNK, and AMPK pathways, are deregulated in cancers including in gliomas. These pathways are also known to be involved in the regulation of protein synthesis [126,127,128,129,130,131,132]. The involvement of these pathways on the regulation of eIFs and their effect on translation in gliomas will successively be described in the following parts (Figure 2).

#### 3.4.2. PI3K/Akt/mTOR pathway

mTOR, a downstream effector of PI3K/Akt pathway, controls cell growth, survival, and motility through protein, lipid, and nucleotide synthesis regulation in response to environmental conditions. mTOR is a serine/threonine (Ser/Thr) protein kinase which is the core of two protein complexes: mTORC1 and mTORC2 [133,134,135,136]. mTORC1 and mTORC2 activities are high in different types of cancer including malignant gliomas. mTORC1 modulates the activity of 4E-BP1 and S6K to control protein synthesis and cell growth. 4E-BP1 phosphorylation through mTORC1 leads to 4E-BP1/eIF4E dissociation. As previously described, free eIF4E can thereby bind the cap structure and eIF4G, and promote eIF4F complex formation leading to protein synthesis. mTORC1-mediated S6K activation modulates the phosphorylation status of several substrates involved in protein synthesis including eIF4G, eIF4B, and PDCD4 leading to protein synthesis activation. In contradiction with data obtained for p-70S6K1, levels of p-4E-BP1 correlate with glioma grade and with patient overall survival, making it a potential prognosis factor to select patients who might benefit from mTOR inhibitor therapies [137]. mTORC2 associates with ribosomes to regulate cell cytoskeleton reshuffle and metabolism [138]. In addition to its well-known function, mTORC2 activates Akt, a kinase also largely implicated in cancers and protein synthesis.

eIF4E/4E-BP1 association controlled by PI3K/Akt/mTOR pathway is a critical step in the translation initiation making this pathway an interesting target for gliomas treatment. For example, Enzastaurin blocks Akt phosphorylation and thus represses Pi3K/Akt pathway, decreases 4E-BP1 phosphorylation, and consequently the formation of the eIF4F complex, leading to apoptosis in GBM cells [139]. In addition, N1,N11-diethylnorspermine (DENSP) which targets the polyamine pathway, has been shown to decrease the expression of mediators of mTOR pathway (i.e., Akt, p-Akt, mTOR, p-mTOR, p-70S6K1, p-p70S6K1, 4E-BP, and p-4E-BP) in GBM cells and to reduce mTOR dependent protein synthesis [140]. mTORC1 is the direct target of rapamycin, an antifungal, immunosuppressive, and antitumor agent. However, rapamycin treatment presumably also inhibits the negative feedback from S6K1 to insulin receptor substrate 1 (IRS1) leading to Akt activation [141]. mTORC1 inhibition with rapamycin analogues (i.e., rapalogs) monotherapy therefore increases Akt-eIF4E pathway and generates therapeutic resistance which has already been widely studied [129,142]. That is why scientists have combined rapamycin analogues with drugs targeting alternative pathways of protein synthesis. For example, Genistein and Biochanin A, two isoflavones used as chemopreventive drugs, block tyrosine kinase receptor (TKR)-Akt pathway and eIF4E phosphorylation in order to sensitize U87MG cells to rapamycin [142]. The association of eEKi-785, a TKR inhibitor, with rapamycin also increases eIF4E-4E-BP1 binding and therefore decreases growth capacity of GBM cells in vitro [143]. Drug combinations targeting the PI3K/Akt/mTOR pathway alter cell growth capacity in vitro and in vivo through eIF4E-4E-BP1 regulation. However, direct consequences of drug combinations on global protein synthesis remain to be demonstrated.

#### 3.4.3. MAPK/MNK Pathway

MNKs (see above), downstream effectors of MAPK pathway, are Ser/Thr kinases able to phosphorylate eIF4E to promote mRNA translation initiation [22]. Interestingly, the two members of the MNK family, MNK1, and MNK2, are overexpressed in GBM [126,144]. MNK1 and 2 share a lot of similarities. They are both able to phosphorylate eIF4E even if they are not equally sensitive to the recruitment by the MAPK pathway [126,145]. Once activated, MNKs associate with the C-terminal domain of eIF4G in order to phosphorylate eIF4E on Ser209 [43,146]. This eIF4E phosphorylation increases its affinity for the m7GTP cap of mRNA. As a consequence, eIF4E deregulation by MNKs promotes cancer cell proliferation, malignant transformation and metastasis [19,41,147]. Interestingly, the constitutive activation of MAPK pathway in GBM induces the phosphorylation of MNK1 and its activation [146,148]. p-eIF4E and p-MNK1 overexpression was also associated with a decreased overall survival of patients with astrocytoma [19].

In line with these observations, MNKs could thus be relevant targets for glioma treatments. Recently developed selective MNKs inhibitors can now be used as potential anticancer therapy. For example, Merestinib inhibits several protein kinases including MNKs, leading to decreased p-eIF4E and increased overall survival of GBM xenograft mice [144]. Other inhibitors of MNKs activity, like CGP57380 and Cercosporamid, have also been correlated with decreased eIF4E phosphorylation, cell cycle arrest, and increased sensitivity of GBM cells to TMZ [149,150]. Silencing of MNK1 with shRNA in U87MG cells decreases their tumorigenicity in a glioma xenograft mouse model [126]. It has also been demonstrated that rapamycin, an mTORC1 inhibitor, can upregulate MNKs pathway and confers resistance to this therapy [129]. The combination of MNKs and mTORC1 inhibitors further inhibits 4E-BP1 phosphorylation at Ser65, increases eIF4E/4E-BP1 association and inhibits glioma cell protein synthesis and proliferation [129]. To illustrate the importance of MNKs pathway in gliomas, a drug called Arsenc trioxide (ATO) (used and approved by the FDA for the treatment of acute promyelocytic leukaemia (APL) with t(15;17) translocation) demonstrated resistance in patient-derived xenograft model of GBM. ATO resistance correlates with higher MNK1 kinase activity and mRNA translation through the MNK/eIF4E pathway. GBM stem cells belonging to the mesenchymal subtype are resistant to ATO and the use of MNK1 inhibitor in combination with ATO could potentially sensitize them to the treatment [127].

#### 3.4.4. AMPK Pathway

AMP-activated protein kinase (AMPK), a highly conserved Ser/Thr kinase, is a sensor of cellular energy allosterically regulated by intracytoplasmic AMP concentration. In conditions of nutrient deprivation, intracellular AMP/ATP ratio increases and activated AMPK is then able to inhibit mTORC1. AMPK is implicated in tumor development and in the regulation of protein synthesis through modulation of mTOR pathway [130,151,152]. AMPK activation by flavones such as Hispidulin and Wogonin as well as other drugs already under clinical investigation like Metformin and 5-aminoimidazole-4-carboxamide ribonucleotide (AICAR) suppresses the mTOR pathway and decreases p-4E-BP1 causing apoptosis and cell cycle arrest in GBM cell lines [153,154,155,156,157,158,159,160]. Wogonin also induces GBM cell apoptosis by inducing ER stress during which protein synthesis is impaired and this implication will be discussed in the next section [109]. Regarding Metformin and AICAR, their modes of action are however not totally dependent on AMPK inhibition [160,161]. They can indeed block cell cycle through direct inhibition of mTOR and by increasing cdc25c phosphatase degradation, respectively [161]. The role of cdc25 phosphatase is to dephosphorylate various cyclin-dependent kinases during cell cycle, allowing its progression. GBM cell apoptosis mediated by AMPK activation is nevertheless in conflict with the observation of the constitutively active status of AMPK in GBM and with the anti-glioma effect of Compound C, also known as dorsomorphin, through its capacity to inhibit AMPK activity [162]. Inhibition of AMPK caused by Compound C is nonetheless not the only event responsible for GBM cell death [161,163]. This discrepancy in the effect of activation or inhibition of AMPK on GBM cells may be the consequence of multiple roles for AMPK in tumors. Indeed, active AMPK inhibits protein synthesis and lipogenesis thereby interfering with tumor growth [157]. AMPK also promotes metabolic reprogramming of cancer cells undergoing metabolic stresses, leading to cancer survival and progression despite an unfavorable environment [164].

## 4. IRES-Dependent Initiation

In addition to the canonical cap-dependent mode of translation (reviewed above), the IRES-dependent mode of initiation has gained interest in the recent years, in the research field of cancer in particular. IRES, first discovered in viruses, consists of RNA secondary structures present in the 5’ UTR end of mRNAs, upstream of the AUG start codon. IRES allows the association of the 40S small ribosomal subunit with mRNA through interaction with trans-activator factors (ITAFs), thus triggering the initiation of translation in a cap-independent manner [17]. ITAFs can be classified into three categories: class I includes ITAFs localized in the nucleus which are able to translocate to the cytoplasm, class II ITAFs are present only in the cytoplasm and class III are non-coding RNA (Table 2) [17]. IRES structures were later discovered in cellular mRNAs and have since been postulated to play a role in cancer development [14]. The first eukaryotic cellular IRES was discovered in the mRNA coding for GRP78 or BiP, a key player in the UPR pathway [165]. Additional cellular IRES have since been identified in mRNA encoding proteins involved in major biological processes such as c-myc, cyclin D1, EGFR, and c-jun [28,124,125,166,167]. Furthermore, a role for IRES in cancer development and resistance has been demonstrated and there is evidence that targeting the IRES machinery could be used as a therapeutic approach for the treatment of cancers, including gliomas [168].

As already discussed above, the mTOR pathway is often altered in gliomas and has led to the development of mTOR inhibitors as a therapeutic approach. Resistance to mTOR inhibitors used as monotherapy is related to the degree of Akt activity. Cancer cells with high Akt activity are sensitive to the mTORC1 inhibitor rapamycin, whereas cancer cells with low Akt activity are resistant. Unfortunately, the majority of GBM cases have low Akt activity and are thus likely to be resistant to mTORC1 inhibition [169]. One of the mechanisms used by GBM cells to resist mTOR inhibitors involves an increase in IRES-dependent translation and thus an upregulation of proteins coded by IRES-containing mRNAs such as c-myc and cyclin D1, two well-known proto-oncogenes [28,124,125]. The treatment of GBM cells presenting low Akt activity with rapamycin activates p38 which in turn triggers IRES mediated translation of c-myc and cyclin D1. Inhibition of p38 genetically, using siRNA, or chemically, prevents rapamycin induced c-myc and cyclin D1 expression. Furthermore, combining mTORC1 and p38 inhibitors significantly enhances G1 cell cycle arrest, increases apoptosis, decreases proliferation in GBM cells in vitro, and inhibits tumor growth in vivo [28]. In addition, Akt activation leads to phosphorylation of hnRNPA1, an ITAF belonging to Class I, which subsequently leads to the inhibition of IRES-dependent translation. Using immunohistochemical and western blot analyses on GBM samples (*n* = 22), they found that elevated Akt activity correlated with increased hnRNPA1 phosphorylation levels [125]. Furthermore, they identified a novel compound able to block the association between hnRNPA1 and its IRES structure localized on c-myc and cyclin D1 which therefore blocks c-myc and cyclin D1 translation and consequently sensitizes GBM cells to mTOR inhibitors. Combining IRES and mTOR therapies significantly decreases GBM cell proliferation in vitro, reduces tumor size in vivo, and increases overall survival in mice grafted with GBM cells [124]. Together, these findings therefore suggest that combined therapy targeting mTORC1 and IRES dependent translation might be a suitable approach for the treatment of GBM [28,124]. Another example highlighting the importance of IRES-dependent translation in gliomas was published by Blau et al. in 2012. They found that c-Jun, a transcription factor often increased in cancer, was overexpressed in GBM at protein levels but not at the level of transcription. They excluded that higher levels of c-jun resulted from accumulation of the protein and more importantly, they demonstrated that this change in c-jun expression was due to an increase in translation in an IRES-dependent manner [166]. Furthermore, the epidermal growth factor receptor (EGFR) is often mutated and overexpressed in cancers including in GBM where increased wild type EGFR can often be observed. Hypoxia, present in GBM microenvironment, is an inhibitor of cap-dependent translation through phosphorylation of eIF2α and an activator of IRES-dependent translation. EGFR protein expression is increased under hypoxic conditions without changes in mRNA levels. This can be explained by the presence of an IRES structure in the 5′ UTR region of EGFR mRNA which allows its translation in the presence of eIF4A acting as an ITAF (see Table 2) [167].

In addition to hnRNPA1 studied by Holmes et al., other ITAFs have been identified in recent years and have been reviewed [17]. ITAFs have several roles aside from regulating IRES-dependent translation. Interestingly, some ITAFs are known to also act as eIFs (Table 2). Moreover, several ITAFs can act as RNA binding proteins (RBP) thereby modulating mRNA stability and translation of different transcripts depending on cell needs. This additional step in the control of mRNA translation will not be described in this review as our focus is the initiation step of translation. We therefore encourage readers to refer to review from Wurth L. and Gebauer F. (2015) for more information about the roles of RBPs in cancers [170]. As part of our work, we have investigated the expression of ITAFs reviewed by Godet et al. (2019) in gliomas using REMBRANDT and TCGA datasets and found that the expression of the majority of ITAFs are altered in gliomas with both increases and decreases being observed (Table 2). According to the REMBRANDT dataset, 78% of investigated ITAFs are overexpressed, 14% are downregulated, and 7.8% are unchanged in gliomas relative to controls. When focusing on GBM subtypes using TCGA database, we found that only 54.5% of ITAFs are increased in GBM tissue whereas 18.2% are significantly decreased and 27.3% of ITAFs are not altered. Altogether, these results suggest an overall overexpression of ITAFs in gliomas which is in agreement with current literature supporting a role for cap-independent translation in cancer including gliomas [168,171]. Additional ITAFs are likely to be identified in the future. In addition, most ITAFs discovered so far have been identified as activators of IRES-dependent translation; however, further work is still required to better understand the role of each ITAF and their potential interaction to unravel their possible implication in glioma development and progression. This is particularly true and important for ITAFs which are also involved in the cap-dependent translation. For example, eIF5B, mostly studied for its role in assembling the small and large ribosomal subunits during cap-dependent translation, has been found to be necessary for the translation of IRES containing mRNAs during cellular stress [172]. eIF5B therefore plays an activating role in cap-dependent and IRES-dependent translation. eIF5B is often overexpressed in cancer which is in agreement with our REMBRANDT data analysis showing that *EIF5B* is overexpressed in gliomas (Table 2) [172]. Interestingly, silencing *EIF5B* in GBM cells stimulates the expression of XIAP, a protein involved in apoptosis which includes an IRES motif in its mRNA. Consequently, suppression of *EIF5B* in GBM cells increased their sensitivity to apoptosis through caspase activation [172]. Other examples of eIFs which have been found to be involved in IRES-dependent translation are eIF4G1 and eIF4G2 where the latter is also known as DAP5. During apoptosis, caspases cleave eIF4G1 into fragments which blocks cap-dependent translation. eIF4G1 cleavage releases a smaller fragment containing the eIF3-eIF4A-binding domain (m4G) which is able to drive IRES-dependent translation. The N-terminal extremity of DAP5 is similar to the m4G domain and can therefore also activate IRES-dependent translation [42]. Indeed, using a cell-free in vitro system, Hundsdoerfer et al. demonstrated that recombinant DAP5 could strongly induce the expression of proteins in an IRES-dependent manner [42]. Interestingly, our REMBRANDT analysis showed that both *EIF4G1* and *EIFG2* are overexpressed in gliomas (Table 2).

IRES are also found in circular RNAs (circRNAs). CircRNAs were first thought to be evolutionary conserved non-coding RNAs; however, it has recently been proven that circRNAs have coding capacity in vitro and in vivo [173,174]. Interestingly, Zhang and colleagues have identified a new circRNA coding for SHPRH146aa, a shorter form of the tumor suppressor SHPRH (a SNF2, histone linker, PHD-finger, RING-finger, and helicase domain-containing protein), a protein containing domains characteristic of DNA repair proteins and transcription factors. Their findings support a role for SHPRH146aa in protecting full length SHPRH from degradation. Interestingly, SHPRH146aa was found to be downregulated in 81% of GBM cases studied [175]. This therefore suggests that expression of IRES-containing mRNA can also be downregulated in GBM. Consequently, developing specific inhibitors targeting particular ITAFs rather than developing global IRES inhibitors might be more suitable for the establishment of novel polytherapy for gliomas. Current strategies developed to target IRES include, among others, antisense oligonucleotides, short hairpin RNAs, small interfering RNAs, and small molecule inhibitors. Their therapeutic potential as well as their respective advantages and disadvantages have been previously reviewed [168].

## 5. Conclusions

This review describes translation initiation occurring in cap-dependent and IRES-dependent manners and assembles the current knowledge on these two initiation mechanisms in gliomas. While it is clear that key players involved in both types of translation initiation are abnormally regulated in gliomas, the majority of existing studies are performed in vitro and in vivo, with research in human glioma samples being limited. Using two freely available datasets, we investigated the gene expression of proteins involved in the regulation of cap-dependent and IRES-dependent machineries and identified new proteins altered in gliomas, indicating possible novel targets for the treatment of gliomas with the potential to ultimately improve survival of GBM patients. Future studies should aim at elucidating whether and how global protein synthesis is affected in human gliomas and at identifying which proteins are actively synthesized by cancerous cells. Indeed, mouse models developed to study translation in vivo permitted to identify mRNAs actively translated in transformed cells compared to non-tumoral surrounding cells [176,177]. This RiboTag technique highlighted genes specifically up-translated or down-translated in cancerous compared to non-tumoral cells. It also showed that changes in translation occurring in gliomas affect specific mRNA rather than being an on/off switch in translation. Drugs currently used to modulate protein synthesis target major indirect pathways instead of specific eIFs or ITAFs, thus increasing the risk of therapeutic escape via compensatory mechanisms. Moreover, current therapies have been shown to modulate translation. For example, GIC irradiation induces an increase in eIF4G translation suggesting an induction of cap-dependent translation upon treatment [178]. As GICs are resistant to radio-therapy and are suspected to play a key role in GBM recurrences, targeting specific actors in protein synthesis could help overcome GBM relapses. Furthermore, targeting key players involved in both the cap-dependent and the cap-independent mechanisms of translation such as 4E-BP1 or members of the eIF3 family would therefore present the advantage of inhibiting both modes of translation.

## Figures and Tables

**Figure 1 cells-08-01542-f001:**
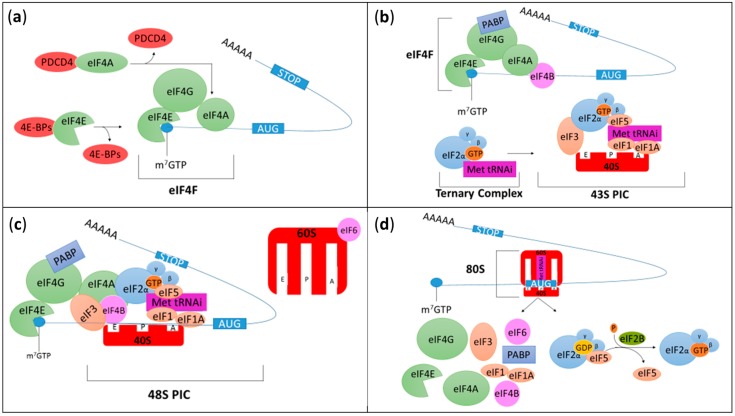
Schematic representation of cap-dependent translation initiation. (**a**) During the first step of translation, the eukaryotic translation initiation factor (eIF) 4F (eIF4F) complex is formed and associates with mRNAs. eIF4F is composed of three subunits (represented in green): eIF4E which binds the 7-methylguanosine 5′-triphosphate (m7GTP) cap structure present on the 5’ end of the mRNA; eIF4G which is a scaffold protein and eIF4A which is an RNA helicase ATP-dependent protein unwinding mRNA during translation. The formation of eIF4F complex can be inhibited by programmed cell death 4 (PDCD4) and eIF4E-binding proteins (4E-BPs) which bind eIF4A and eIF4E, respectively. (**b**) During the second step of cap-dependent translation, the small ribosomal subunit (40S, red) associated with eIF1, eIF1A, eIF3 and eIF5 interacts with the ternary complex constituted of eIF2, a switch-protein bound to guanosine triphosphate (GTP) and the initiator methionyl-transfer RNA (Met-tRNAi). Together they form the 43S preinitiation complex (43S PIC). eIF2 is a heterotrimeric structure composed of three subunits (α, β and γ). At this stage, poly(A) binding protein (PABP, an RNA-binding protein) binds eIF4G and the 3’ poly(A) tail to induce mRNA circularization. eIF4B also interacts with eIF4A to stabilize unwounded mRNA and stimulate eIF4A RNase and helicase activities. (**c**) The 43S PIC associates with the eIF4F complex to form the 48S PIC before scanning the mRNA until it reaches and recognizes the start codon (AUG), and Met-tRNAi binds the peptidyl (P) site of the ribosome. The 60S ribosomal subunit associated with eIF6 will then be recruited. (**d**) The 60S binds the 40S which triggers eIF2-GTP hydrolysis into guanosine diphosphate bound (GDP), leading to eIFs release and the formation of the 80S initiation complex. At this stage, the initiation complex is fastened on the mRNA and the elongation begins. The aminoacyl (A) and the exit (E) sites present in the ribosome correspond to the sites where a second tRNA with its amino acid enters the ribosome and the amino acid depleted tRNA exits the ribosome, respectively. Finally, once the ternary complex has been released, eIF2B frees up eIF5 and restores eIF2-GDP into eIF2-GTP.

**Figure 2 cells-08-01542-f002:**
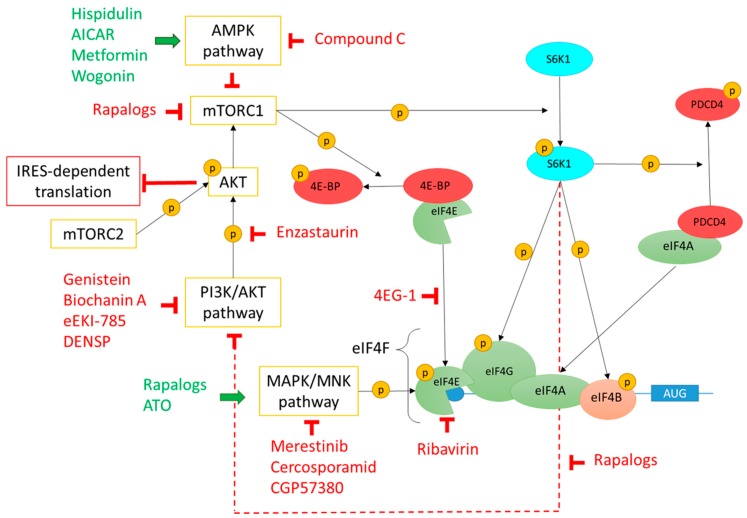
Regulation of eIF4F complex formation by PI3K/Akt/mTOR, MAPK/MNK, and AMPK pathways and drugs targeting these pathways. mTOR, a downstream effector of PI3K/Akt pathway, is a serine/threonine kinase which is the core of two protein complexes: mTORC1 and mTORC2. mTORC1 phosphorylates 4E-BP and the S6K1 kinase. 4E-BP phosphorylation leads to 4E-BP1/eIF4E dissociation and consequently formation of the eIF4F complex. S6K1 phosphorylation leads to its activation which in turn phosphorylates eIF4G, eIF4B, and PDCD4. Once phosphorylated, PDCD4 is degraded thus freeingeIF4A which can then join the eIF4F complex. Phosphorylated eIF4B stabilizes unwounded mRNA and stimulates eIF4A RNase and helicase activities. Phosphorylated S6K1 also inhibits the PI3K/Akt pathway by a feedback loop (dotted red line). mTORC2 can phosphorylate and activate Akt which inhibits IRES-dependent translation. MNKs are downstream effectors of the MAPK/MNK pathway able to phosphorylate eIF4E which then promotes mRNA translation initiation. Finally, the AMPK pathway modulates protein synthesis by inhibiting mTORC1 and consequently blocking 4E-BP/eIF4E dissociation and eIF4F complex formation. Drugs acting as activators or inhibitors of these different pathways are represented in green or in red, respectively. mTOR—mechanistic target of rapamycin; PI3K—phosphoinositide 3-kinase; Akt—protein kinase B; mTORC—mechanistic target of rapamycin complex; 4E-BP—eukaryotic translation initiation factor 4E-binding protein; S6K1—S6 kinase 1; eIF—eukaryotic initiation factor; PDCD4—programmed cell death 4; IRES—internal ribosome entry site; MAPK—Mitogen-activated protein kinase (MAPK); MNK—MAPK interacting protein kinases; AMPK—AMP-activated protein kinase. Phosphorylation is represented by a yellow sphere containing the letter P.

**Table 1 cells-08-01542-t001:** Expression of factors involved in the cap-dependent initiation step in gliomas. mRNA expression levels of eukaryotic initiation factors (eIFs) and other players of the cap-independent initiation detected in control brain tissues (*n* = 21) were compared with their expression in the three glioma subtypes (oligodendroglioma, *n* = 66; astrocytoma, *n* = 145; glioblastoma multiform (GBM), *n* = 214) using the REMBRANDT database. Levels of these mRNAs from control tissues (*n* = 11) were then compared to expression found in the four GBM subtypes defined by the Verhaak’s classification (classical, *n* = 54; menchymal, *n* =58; neural, *n* =33; proneural, *n* = 57) using the TCGA dataset. These two datasets were obtained through the independent Betastasis genomics analysis and visualization platform. GraphPad Prism (version 5.03 for Windows, GraphPad Software, San Diego California USA, www.graphpad.com) was used for statistical analysis. D’Agostino & Pearson omnibus normality test was used to control for normal distribution. One-way analysis of variance (ANOVA) followed by Bonferroni’s Multiple Comparison Test was used for parametric analysis and if required Kruskal-Wallis test followed by Dunn’s Multiple Comparison Test was performed for non-parametric analysis. ns—not significant, +/− *p* < 0.05, ++/− − *p* < 0.01, and +++/− − − *p* < 0.001 where “+” and “−“ indicate an increase and a decrease in expression, respectively. REMBRANDT—REpository for Molecular BRAin Neoplasia DaTa; TCGA—The Cancer Genome Atlas.

eIFs and Modulators	Gene	REMBRANDT			TCGA			
		*Oligodendroglioma*	*Astrocytoma*	*GBM*	*Classical*	*Mesenchymal*	*Neural*	*Proneural*
eIF1	*EIF1*	ns	ns	ns	ns	ns	ns	ns
eIF1A	*EIF1A*	ns	ns	ns	ns	ns	ns	ns
eIF1AD	*EIF1AD*	+ +	ns	+ + +	ns	ns	ns	ns
eIF2 alpha	*EIF2S1*	ns	ns	ns	ns	ns	ns	ns
eIF2 beta	*EIF2S2*	+ +	+	+ + +	ns	ns	ns	ns
eIF2 gamma	*EIF2S3*	+ + +	+ +	+ + +	ns	+	ns	ns
HRI	*EIF2AK1*	+ +	+	+ +	ns	ns	ns	ns
PKR	*EIF2AK2*	+	+ +	+	ns	ns	ns	ns
PERK	*EIF2AK3*	ns	ns	+ + +	ns	ns	ns	ns
GCN2	*EIF2AK4*	ns	ns	+ +	ns	ns	ns	ns
eIF2B1	*EIF2B1*	+ + +	+ + +	+ + +	ns	ns	ns	+ +
eIF2B2	*EIF2B2*	+ + +	+ + +	+ + +	ns	ns	ns	ns
eIF2B3	*EIF2B3*	− − −	− − −	ns	−	ns	ns	ns
eIF2B4	*EIF2B4*	+ + +	+ + +	+ + +	ns	ns	+	ns
eIF2B5	*EIF2B5*	+ +	+	+ +	ns	ns	ns	ns
eIF3A	*EIF3A*	+ + +	+ + +	ns	ns	ns	ns	ns
eIF3B	*EIF3B*	+ + +	+ + +	+ + +	+ + +	+ + +	ns	+ +
eIF3D	*EIF3D*	+ + +	+ + +	+ + +	+	+ +	ns	+ +
eIF3E	*EIF3E*	+ + +	+ + +	+ + +	ns	ns	ns	ns
eIF3F	*EIF3F*	+ + +	+ + +	+ + +	ns	ns	ns	ns
eIF3G	*EIF3G*	+ + +	+ + +	+ + +	+ + +	+ +	+ +	+ +
eIF3H	*EIF3H*	+ + +	+ + +	+ + +	ns	ns	ns	ns
eIF3I	*EIF3I*	ns	ns	+ + +	+	+	+ + +	ns
eIF3J	*EIF3J*	+	ns	ns	ns	ns	ns	ns
eIF3K	*EIF3K*	ns	ns	ns	ns	ns	+	ns
eIF3L	*EIF3L*	+ + +	+ + +	ns	ns	ns	ns	ns
eIF3M	*EIF3M*	+ + +	+ +	+ + +	ns	ns	ns	ns
eIF4A1	*EIF4A1*	+ + +	+ + +	+ + +	+ +	+ + +	+	+ + +
eIF4A2	*EIF4A2*	− − −	− − −	− − −	− − −	− − −	ns	− − −
eIF4A3	*EIF4A3*	+ + +	+ + +	+ + +	+	+ +	+ + +	+ +
eIF4B	*EIF4B*	+ +	ns	ns	ns	ns	ns	ns
eIF4E1	*EIF4E1*	− − −	− − −	− − −	− − −	− − −	ns	− − −
eIF4E2	*EIF4E2*	+ +	+ +	+ + +	+ +	+ + +	+ + +	+
eIF4E3	*EIF4E3*	− − −	− − −	− − −	− − −	− − −	ns	− − −
4E-BP1	*EIF4EBP1*	+ + +	+ + +	+ + +	+	+ + +	+ +	+ + +
4E-BP2	*EIF4EBP2*	+ + +	+ + +	ns	+	+	+	ns
4E-BP3	*ANKHD1*	+ + +	+ + +	+ + +	ns	ns	ns	ns
4E-T	*EIF4ENIF1*	ns	−	− − −	− − −	− − −	− − −	− −
eIF4G1	*EIF4G1*	+ + +	+ + +	+ + +	ns	ns	ns	ns
eIF4G2	*EIF4G2*	+ + +	ns	+ + +	ns	ns	ns	ns
eIF4G3	*EIF4G3*	− − −	− − −	− − −	− − −	− − −	− − −	− −
eIF4H	*EIF4H*	ns	ns	ns	ns	ns	−	ns
PDCD4	*PDCD4*	ns	ns	ns	− −	ns	ns	ns
eIF5	*EIF5*	ns	− −	ns	ns	−	ns	− −
eIF5B	*EIF5B*	+ + +	+ + +	+ + +	ns	ns	ns	ns
eIF6	*EIF6*	+ + +	+ +	+ + +	+ +	+ +	+ +	ns

**Table 2 cells-08-01542-t002:** Expression of IRES trans-acting factors in gliomas. Internal ribosomal entry site (IRES) trans-acting factors (ITAFs) can act as activator (A) or inhibitor (I) of cap-independent translation. Class I ITAFs are localized in the nucleus and can translocate to the cytoplasm; class II ITAFs are only present in the cytoplasm [17]. The expression of ITAFs reviewed by Godet AC et al. (2019) was compared between control brain tissue (*n* = 21) and gliomas (oligodendrogliomas, *n* = 66; astrocytomas, *n* = 145; glioblastoma multiform (GBM), *n* = 214) using the REMBRANDT database. ITAF expression from control tissue (*n* = 11) was then compared to expression found in the four GBM subtypes defined by the Verhaak’s classification (classical, *n* = 54; menchymal, *n* = 58; neural, *n* = 33; proneural, *n* = 57) using the TCGA dataset. These two datasets were obtained through the independent Betastasis genomics analysis and visualization platform. GraphPad Prism (version 5.03 for Windows, GraphPad Software, San Diego California USA, www.graphpad.com) was used for statistical analysis. D’Agostino & Pearson omnibus normality test was used to control for normal distribution. One-way analysis of variance (ANOVA) followed by Bonferroni’s Multiple Comparison Test was used for parametric analysis and if required, Kruskal-Wallis test followed by Dunn’s Multiple Comparison Test was performed for non-parametric analysis. ns: not significant, +/− *p* < 0.05, ++/− − *p* < 0.01, and +++/− − − *p* < 0.001 where “+” and “−” indicate an increase and a decrease in expression, respectively. # indicates ITAFs also acting as eIFs, REMBRANDT: REpository for Molecular BRAin Neoplasia DaTa; TCGA: The Cancer Genome Atlas.

ITAFs	Gene	REMBRANDT			TCGA				
		*Oligodendroglioma*	*Astrocytoma*	*GBM*	*Classical*	*Mesenchymal*	*Neural*	*Proneural*	*Activity*
**Class I**									
Annexin A2	*ANXA2*	ns	+ +	+ + +	+ + +	+ + +	+	ns	A
CUGBP1	*CUGBP1*	ns	ns	ns	ns	− −	− − −	ns	A/I
DAP5 #	*EIF4G2*	+ + +	ns	+ + +	ns	ns	ns	ns	A
FBP3	*FUBP3*	+ + +	+ + +	+ + +	+ + +	+ + +	+	+ + +	A
FUS	*FUS*	+ + +	+ + +	+ + +	+ +	ns	ns	+ + +	A
GRSF1	*GRSF1*	− − −	− − −	− − −	− −	− − −	ns	−	A
H-ferritin	*FTH1*	ns	ns	ns	ns	ns	ns	ns	A
HDMX	*MDM4*	+ + +	+ + +	+ + +	ns	ns	ns	+ +	A
hnRNPA1	*HNRNPA1*	+ + +	+ +	ns	+ +	ns	ns	+ + +	A/I
hnRNPC	*HNRNPC*	+ + +	+ + +	+ + +	ns	ns	ns	ns	A
hnRNPD	*HNRNPD*	+ + +	+ + +	+	ns	ns	ns	ns	A
hnRNPE1	*PCBP1*	ns	ns	+ +	ns	ns	ns	ns	A
hnRNPE2	*PCBP2*	+ + +	+ + +	+ + +	+	ns	ns	+ + +	A
hnRNPE3	*PCBP3*	ns	ns	− − −	− − −	− − −	− −	− −	A
hnRNPE4	*PCBP4*	+ +	ns	ns	ns	ns	ns	+ + +	A
hnRNPH2	*HNRNPH2*	ns	ns	− − −	− − −	− −	− −	− −	A
hnRNPK	*HNRNPK*	+ + +	+ + +	+ + +	+ +	ns	ns	+ + +	A
hnRNPL	*HNRNPL*	ns	ns	ns	ns	ns	ns	+ + +	A
hnRNPM	*HNRNPM*	+ + +	+ + +	+ + +	+ +	ns	ns	ns	A
hnRNPQ	*SYNCRIP*	+ + +	+ + +	+ + +	+	ns	ns	+ +	A
hnRNPR	*HNRNPR*	+ +	+ + +	+ + +	ns	ns	ns	+	A
HuR	*ELAV1*	+	+	+ + +	− −	− − −	− − −	ns	A/I
La auto antigen	*SSB*	+	+	+ +	ns	ns	ns	ns	A/I
Mdm2	*MDM2*	+ + +	+ + +	+ + +	ns	+	+	ns	A
NF45	*ILF2*	+ + +	+ + +	+ + +	+ +	ns	+ + +	+ + +	A
nPTB	*PTBP2*	− − −	− −	− − −	− − −	− − −	ns	ns	A
nucleolin	*NCL*	Not available			ns	ns	ns	ns	A/I
p54nrb	*NONO*	+ + +	+ + +	+ + +	+ + +	+ +	ns	+ + +	A
PDCD4 #	*PDCD4*	ns	ns	ns	− −	ns	ns	ns	A/I
PSF	*SFPQ*	+ +	ns	ns	ns	ns	ns	+	A/I
PTB	*PTBP1*	+ + +	+ + +	+ + +	+ + +	+ + +	ns	+ + +	A/I
RHA	*DHX9*	+ + +	+ + +	+ + +	ns	ns	ns	+	A
SMAR 1	*BANP*	+ + +	+ + +	+ + +	+	+ +	+ + +	+ +	A/I
YB1	*YBX1*	+ + +	+ + +	+ + +	+ + +	+ +	+	+ + +	A
**Class II**									
4E-BP1 #	*EIF4EBP1*	+ + +	+ + +	+ + +	+	+ + +	+ +	+ + +	A
APP (AICD)	*APP*	− − −	− − −	− − −	− −	− −	− − −	− − −	A
eeF1A2	*EEF1A2*	− − −	− − −	− − −	− − −	− − −	−	−	A
eIF3A #	*EIF3A*	+ + +	+ + +	ns	ns	ns	ns	ns	A
eIF3B #	*EIF3B*	+ + +	+ + +	+ + +	+ + +	+ + +	ns	+ +	A
eIF3D #	*EIF3D*	+ + +	+ + +	+ + +	+	+ +	ns	+ +	A
eIF3E #	*EIF3E*	+ + +	+ + +	+ + +	ns	ns	ns	ns	A
eIF3F #	*EIF3F*	+ + +	+ + +	+ + +	ns	ns	ns	ns	A
eIF3G #	*EIF3G*	+ + +	+ + +	+ + +	+ + +	+ +	+ +	+ +	A
eIF3H #	*EIF3H*	+ + +	+ + +	+ + +	ns	ns	ns	ns	A
eIF3I #	*EIF3I*	ns	ns	+ + +	+	+	+ + +	ns	A
eIF3J #	*EIF3J*	+	ns	ns	ns	ns	ns	ns	A
eIF3K #	*EIF3K*	ns	ns	ns	ns	ns	+	ns	A
eIF3L #	*EIF3L*	+ + +	+ + +	ns	ns	ns	ns	ns	A
eIF3M #	*EIF3M*	+ + +	+ +	+ + +	ns	ns	ns	ns	A
eIF4A1 #	*EIF4A1*	+ + +	+ + +	+ + +	+ +	+ + +	+	+ + +	A
eIF4A2 #	*EIF4A2*	− − −	− − −	− − −	− − −	− − −	ns	− − −	A
eIF4A3 #	*EIF4A3*	+ + +	+ + +	+ + +	+	+ +	+ + +	+ +	A
eIF4G1 #	*EIF4G1*	+ + +	+ + +	+ + +	ns	ns	ns	ns	A
eIF5B #	*EIF5B*	+ + +	+ + +	+ + +	ns	ns	ns	ns	A
eL38	*RPL38*	+ + +	+ + +	+ +	−	ns	ns	ns	A
eS19	*RPS19*	+ + +	+ + +	+ + +	+ + +	+ + +	+ + +	+ +	A
eS25	*RPS25*	+ + +	+ + +	+ + +	ns	ns	+ +	ns	A
Gemin5	*GEMIN5*	+ + +	+ + +	+ + +	+	+	ns	+ +	A/I
Hepsin	*HPN*	− − −	− − −	−	ns	ns	ns	ns	I
PINK1	*PINK1*	− − −	− − −	− − −	− − −	− − −	− −	− − −	A
Rack1	*GNB2L1*	+ + +	+ + +	+ + +	+ + +	+ + +	+	+ + +	A/I
TCP80	*ILF3*	+ + +	+ + +	+ + +	+ + +	ns	ns	+ + +	A
uL1	*RPL10A*	+ + +	+ + +	+ + +	ns	ns	+ +	ns	A
uL24	*RPL26*	Not available			ns	ns	ns	ns	A
uL5	*RPL11*	+ + +	+ + +	+ + +	ns	+ + +	+	+ + +	A
VASH1	*VASH1*	+	+ +	+	+ +	ns	ns	+ +	A

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
