# Peer review of "Relevance of Translation Initiation in Diffuse Glioma Biology and its Therapeutic Potential"

_cells, 2019, doi:10.3390/cells8121542_

Round 1

Reviewer 1 Report

Marina and colleagues present an excellent review of the literature concerning gliomas and the therapeutic potential of targeting translation. 

Minor concerns:

The review lacks any mention of the cap-dependent translation inhibitor, pateamine A, which targets eIF4A and has demonstrable in vivo anti-cancer activitiy [Bordeleau et al. PNAS 2005; Low et al. Mol. Cell 2005; Low et al. Meth Enzy 2007; Kuznetsov et al. Mol. Can. Ther. 2009

Table legends contain needlessly duplicative text on methodology, for example the information on graphpad and the n values.

Please define abbreviations when first used: WHO in line 33.

Hyphens are needed in lines 41 and 42: "astrocyte cell-like"

Space missing between expression and initiation in line 208.

First sentence of conclusion is not well constructed with parallelism:
"review describes ... and assembled" should be "review describes ... and assembles".  While the last sentence of the conclusion "both mode of translation" should be "both modes of translation." I recommend a more careful proofreading by a qualified native English speaker.

Reviewer 2 Report

In this review, Digregorio and colleagues provide a complete and well-documented analysis of the implication of translation initiation mechanism in diffuse glioma biology. The review includes a detailed presentation of the translation initiation mechanism for cap-dependent translation, subdivided for each initiation factor. In addition, the review provides a long section on IRES-driven translation and associated regulatory RNA-binding proteins. Finally, Digregorio et al. include the analysis of REMBRANDT and TCGA RNA expression datasets for gliomas and calculate relative expression levels for regulatory and initiation factors of translation initiation.  Overall, the review is clearly written and would be of importance for researchers in the field of glioma with interest in protein synthesis.

Nevertheless, some improvements and corrections of the present manuscript are absolutely needed. These include careful examination and correction of cited references (including, but not limited to, the details below), and inclusion of relevant missing-references for omics analysis and related mouse model. Some paragraphs rearrangement would be necessary to avoid redundancy. Finally, discussion on therapeutic potential is somehow limited.

General comments:

The authors have discussed the role of initiation and regulatory factors but forgot to mentioned omics analysis of translation. Translatome analyses (riboseq or poysome profiling) demonstrate translational dysregulation and identity involved mRNAs. Of interest Ribosome Profiling from Gonzalez and colleagues in J. Neuroscience 2014 as well as poysome profiling in Wahba et al. in Cancer Res 2008 will be a good start together with Helmy et al . in Plos One 2012. The article form Gonzalez et al. would also permit the introduction of a superb mouse model to study translation in glioma, the RiboTag.

I would suggest to regroup the ER stress paragraph with the integrated stress response paragraph (AKA the 4 eIF2A kinases). Considering the objective of the review on translation, PERK (eIF2AK3) is the only arm of UPR controlling translation initiation. ATF6 and IRE1 are irrelevant to this review.

The presentation of data analysis of REMBRANDT and TCGA RNA expression is far from being easy to read, especially rows title identifying brain tumor subtypes (table 1 and 2). In addition, the analysis highlight opposite findings in the two database. Did the authors check for proper assignment of tumors subtypes in the TCGA database, some errors have been reported (and corrected) for pancreatic cancers (Nicolle R et al Cancers 2019). This will avoid biased conclusions.

The list of ITAFs provided in table 2 includes numerous RNA binding proteins. RBPs have known functions in regulating mRNA translation aside from IRES-driven translation. This should be corrected, or at least discussed (See Moore Front Physiol 2018 or Wurth L. and Gebauer F BBA 2015 for review)

Detailed comments:

eIF4E2 is also called 4EHP.

Line 143: Ref 22 is wrong. It doesn’t describe interaction of the 4Es to the cap but the binding to 4E-T.

Line 158: Phosphorylation of eIF4E and affinity to the cap have been subjected to debate. As eIF4E needs to bind the cap and stabilized by eIF4G in order to be phosphorylated. It’s more commonly accepted that phosphorylated eIF4E modifies the repertoire of translated mRNA (Ref on 4E-KI animals Furic PNAS 2010 Robichaud Oncogene 2015). Same on Line 178

Line 164: Refs 25 and 26 are not about Glioma. Ref 25 is a review on mRNA translation. Ref 26 is an article on lymphoma.

Line 173: Should mentioned that phosphorylation of PDCD4 leads to its degradation (Schmid Cancer res 2008) rather than released of eIF4A.

Line 191: Same as line 143. Must cite Rosettani P. J. Mol Biol 2007 in addition to Joshi B Eur. J. Biochem. 2004.

Line 218: ref 31 is a review and doesn’t refer to gliomas per se, as do refs 29 and 30. In addition, Ref 29 and 30 only infer the role of eIF4E without any clear demonstration. Should rather cite general example from N. Sonenberg or D. Ruggero lab.

Line 219-220: Ref 33 is about ribavirin not about eIF4E per se. Sentence is not meaningful (supported by …that eIF4E supports…).

Line 242: ref 40 is a review. The proper reference is Dostie et al. Embo J 2000.

Line 253: Dap5, the other name of eIF4G2, should be mentioned here.

Line 257: add ref 122.

Line 264-267: Ref 46 is about eIF4G2 (aka Dap5) not eIF4G3. There is no such thing as protein translation, only mRNA is translated. Proteins are synthetized (same at line 375).

Line 273: ref 47 is about 4Gi-1 not about 4E-BPs. Proper ref would be from Gingras and Sonenberg Genes & Dev 2001 or 1999

Line 282 and 287: Same as line 164.

Line 285: Silenced is more adapted than loss, as the paper uses siRNA.

Line 294: Ref 51 is not the most relevant, should add Parsyan Nat Rev Cell Biol 2011 or Chu and Pelletier BBA 2014 for review.

Line 299: Only ref 52 and 53 are relevant.

Line 317: Ref 59 should be removed. It doesn’t provide direct evidence of the role of PDCD4.

Line 325: The excellent review from Hershey in BBA 2015 must be mentioned.

Line 588-591: Perk controls the translation of ATF4 mRNA but doesn’t “activate” ATF4.

Line 640:  Yanq & Sarnow Nucl Acid Res 1997 should be added.

Round 2

Reviewer 2 Report

I would like to congratulate authors for the improvement of this review on translation initiation in diffuse glioma biology, which is now suitable for publication.

I hope that the authors have found my comments constructive.